# Tackling Surgical Morbidity and Mortality through Modifiable Risk Factors in Cancer Patients

**DOI:** 10.3390/nu14153107

**Published:** 2022-07-28

**Authors:** Boram Lee, Ho-Seong Han

**Affiliations:** Department of Surgery, Seoul National University Bundang Hospital, Seoul National University College of Medicine, Seoul 13620, Korea; boramsnubhgs@gmail.com

**Keywords:** postoperative complications, malnutrition, inflammation, minimally invasive surgical procedures, neoplasms

## Abstract

Despite advances in surgical techniques, surgical morbidity and mortality remain important public health problems. Postoperative complications often lead to greater morbidity and mortality, as well as increased length of hospital stay and medical costs. Therefore, a reduction in postoperative complications is particularly important with regard to positive long-term outcomes in patients with cancer. To improve patients’ postoperative prognosis, it is necessary to screen for and focus on modifiable risk factors and their subsequent resolution. Recently, it was reported that nutritional status, inflammation and surgical approaches are related to postoperative morbidity and mortality. Therefore, in this review article, we describe the current evidence regarding modifiable risk factors influencing surgical morbidity and mortality as well as future directions for improved postoperative management in cancer patients.

## 1. Introduction

Despite advances in surgical techniques, surgical morbidity and mortality remain pressing public health problems. Postoperative complications range from 7% to 50% depending on the surgery type, patient risk factors, and the definition of complications [1,2,3]. They are often strongly correlated with increased morbidity and mortality, as well as a greater duration of hospital recovery and healthcare costs [4]. Recent data have suggested that patients who suffer from morbidity have poorer disease-free and long-term survival rates [5,6,7]. Additionally, infectious complications, such as intra-abdominal abscess, abdominal infection, and pneumonia, may reflect immune suppression and can also lead to cancer recurrence [8,9,10]. Furthermore, postoperative morbidity can also delay initiation of adjuvant chemotherapy post-surgery, which may further affect disease prognosis [10]. Therefore, the prevention of postoperative complications is essential for positive long-term outcomes, especially in patients with cancer [4].

The identification of risk factors provides an opportunity to optimize perioperative care through improved patient management and a diminished risk of postoperative complications and associated mortality. Various clinical factors are known to influence postoperative morbidity and mortality: (i) Patient characteristics: age, frailty, male sex, and the American Society of Anesthesiologist (ASA) score; (ii) preoperative inflammatory biomarkers, (iii) comorbidities; (iv) intoxication: smoking and alcohol consumption; (v) nutritional-related risk factors: malnutrition/preoperative weight loss and sarcopenia; (vi) disease-related risk factors: preoperative tumor complications and advanced tumor stage; and (vii) intraoperative techniques and care. Among these clinical factors, tumor status, surgery difficulty, and comorbidities are intrinsic to a patient and cannot be directly addressed. Therefore, to improve the patient’s postoperative prognosis, it is necessary to focus on modifiable risk factors, which can be partially or fully treated. Recently, it has been reported that the nutritional status, inflammation, frailty and surgical approach are related to postoperative morbidity and mortality (Figure 1). Therefore, in this review article, we summarize the current evidence regarding these modifiable risk factors influencing surgical morbidity and mortality in cancer patients.

## 2. Methodology

A systematic search of the literature related to “nutrition” or “inflammation” or “minimally invasive surgery” and “cancer” was conducted in PubMed advanced search builder, Embase, Web of Science and Cochrane database for publications before May 2022. The reference lists and all included studies were manually verified for further additional eligible studies. After excluding duplications, the titles and abstracts of remaining records were independently selected by two authors (H-S Han, B Lee).

## 3. Nutrition

Malnutrition is a prevalent problem among cancer patients, and occurs in approximately 20% to more than 70% of the patients. A more advanced stage of malnutrition leads to cancer anorexia–cachexia syndrome (an over catabolic condition characterized by weight loss and sarcopenia), which occurs in about 15% to about 40% of cancer patients [11]. Malignancy-associated malnutrition is caused by various alterations. Firstly, inadequate nutritional intake is frequently observed in patients with cancer, which often results in severe weight loss. Secondly, muscle protein depletion is a hallmark of cancer cachexia, immensely impinging patient quality of life and negatively impacting physical function and treatment tolerance. Furthermore, systemic inflammation syndrome is frequently activated in patients undergoing chemotherapy. This can vary in degree but impacts all relevant metabolic pathways [12]. Finally, malnutrition is associated with decreased functioning of the immune, respiratory, and cardiac systems, and is known to hinder the overall healing process [13]. For these reasons, malnutrition greatly influences short-term and long-term outcomes post-surgery.

### 3.1. Preoperative Nutritional Status and Its Influence on the Outcomes of Surgery

Malnutrition is a well-established risk factor for postoperative morbidity and mortality in surgical patients, with previous studies reporting adverse outcomes after surgery. Studies also show that perioperative fasting, surgical trauma, and postoperative stress can exacerbate malnutrition [13]. Malnourished patients who underwent surgery had greater postoperative complications and longer hospital stays than patients without preoperative malnutrition [13,14,15,16]. The loss of skeletal muscle and visceral proteins is considered an important indicator of nutritional status [17]. The depletion of muscle and protein leads to changes in electrolyte regulation and decreased metabolic activity. These adverse bodily changes coexist with immunosuppression, an important mechanism of increased postoperative complications [18]. Han et al. also reported that preoperative malnutrition is associated with increased blood loss during surgery [16], which is believed to be due to hypoalbuminemia caused by malnutrition.

Preoperative malnutrition can also affect poor long-term outcomes [19]. Han et al. found that preoperative malnutrition was significantly associated with a diminished long-term overall survival, disease-free survival and cancer specific survival rates for patients with pancreatic cancer [20]. It is still unclear why the preoperative malnutrition has an adverse effect on survival, although several possible explanations exist. First, malnutrition may be associated with immunosuppression, which may provide a favorable microenvironment for tumor recurrence, which may lead to recurrence after surgery [21,22]. Additionally, the presence of malnutrition can reduce the tolerance and responses to treatment, which can also lead to a poor prognosis in cancer patients [23,24]. Furthermore, malnutrition patients may have a higher risk of non-cancer death. Migita et al. [25] found that underweight patients with gastric cancer were more likely to die from non-cancerous causes, particularly infection, than normal-weight patients with gastric cancer. Therefore, early recognition and prompt intervention to optimize the perioperative status of patients scheduled to undergo surgery should be an integral part of perioperative assessments. The improvement of surgical outcomes depends not only on the improvement of surgical skills but also the promotion of nutrition management.

### 3.2. Nutritional Risk Screening and Assessment

Nutritional risk screening tools are very useful for detecting potential or overt malnutrition in a simple and concise manner. Diverse scores and screening systems have been established in recent decades for use in various clinical setting and patient populations [26]. Screening should be carried out within the first 24–48 h after hospital admission and regularly thereafter. The European Society for Clinical Nutrition and Metabolism (ESPEN) recommends the Nutritional Risk Screening 2002 (NRS-2002) [27] in an inpatient setting, the Malnutrition Universal Screening Tool (MUST) [28] for an ambulatory setting, and the Mini Nutritional Assessment (MNA) [29] for institutionalized patients [30].

Nutritional assessment should be performed on patients identified as at risk of nutritional risk following the first step. This assessment allows the clinician to gather more information and conduct a nutrition-focused physical examination in order to determine both the presence and severity of an underlying nutritional deficit [31]. The Patient-Generated Subjective Global Assessment (PG-SGA) is a nutritional assessment tool for oncology practice and research [32].

Despite most definitions of malnutrition that include similar risk factors, there is a lack of consensus on the diagnostic criteria for applications in clinical settings. Therefore, the Global Leadership Initiative in Malnutrition (GLIM) recently published a new definition of malnutrition for adults, based on a two-step model for risk screening and diagnostic assessment [33]. These definitions are based on both phenotypic criteria (weight loss, low body mass index (BMI) and reduced muscle mass), and etiologic criteria (reduced food intake or assimilation and inflammation). Although such conditions are thought to be prevalent among patients with gastrointestinal disease, the frequency and severity of malnutrition in patients with gastrointestinal disease is not well-documented. Moreover, the GLIM also encourages the nutrition community to use criteria both in prospective and retrospective cohort studies as well as trials to validate relevance in clinical practice [33]. 

### 3.3. Nutritional Support

Nutrition care should be an integral part of the multimodal treatment for cancer patients to improve surgical patient outcomes. A meta-analysis concluded that short-term (7~10 days) preoperative nutritional therapy should be considered for patients with mild malnutrition. In severely malnourished patients, nutritional control and light aerobic exercise should be combined for nearly 15 days [34]. The 2017 ESPEN guidelines strongly recommend nutritional intervention prior to major surgery in case of significant nutritional risk or apparent malnutrition for at least 10 to 14 days, even at the cost of deferring surgical intervention [30]. However, postponing surgery for nutritional support in cancer patient can also negatively affect patient’s survival. Therefore, it is important to determine the optimal timing and duration of preoperative nutrition [20].

## 4. Inflammation

Inflammation is the body’s response to tissue damage, and can occur from physical or ischemic injury, infection, toxin exposure, or other types of trauma. The body’s inflammatory response triggers cellular changes and immune responses that repair damaged tissue and cause cell growth in the damaged tissue area. Inflammation can become chronic when it persists or there is a failure in immune regulation. When these inflammatory responses become chronic, cellular mutations and proliferation occur, often creating a thriving environment for cancer development [35]. The preoperative inflammatory activity is related to a higher risk of postoperative infectious complications [36]. Many studies also suggested that the preoperative inflammation could predict the long-term outcomes of patients with various tumors. Therefore, treating the underlying cause of inflammation is vital to ensuring a positive post-surgical outcome.

### 4.1. Prognostic Effect of the Preoperative Inflammation on Short-Term Outcomes 

Preoperative persistent inflammation is associated with an increased risk of postoperative infectious inflammation and intra-abdominal infection [37,38]. The mechanism linking preoperative and postoperative inflammation remains to be determined. It has been hypothesized that the enhancement of the systemic inflammatory response may be involved [39]. Preoperative inflammation can lead to hematopoietic changes (including alterations in the relative number of leukocytes), changes in acute-phase reactions (including an elevation in C-reactive protein level T and a decline in serum albumin level), alterations in the neuroendocrine system, and changes in protein and energy metabolism [40]. These physiological changes may lead to an increased risk of postoperative infectious complications or surgical site infections in cancer patients [41]. Further studies supporting the role of inflammation in the onset of postoperative infectious complications come from clinical trials that showed the benefit of preoperative anti-inflammatory drugs in preventing infectious complications [42,43]. Advances in morbidity were shown when anti-inflammatory agents were used preoperatively and immediately after operation [42,43]. Srinivasa S et al. [42] concluded that the preoperative administration of glucocorticoid decreases complications and length of hospital stay after major abdominal surgery and is a likely consequence of attenuating the postsurgical inflammatory response. STARSurg Collaborative group [43] found that the early use of non-steroidal anti-inflammatory drugs (NSAIDs) is associated with a reduction in postoperative adverse events following major abdominal surgery.

### 4.2. Prognostic Effect of the Preoperative Inflammation on Long-Term Outcomes

Inflammation promotes carcinogenesis by inducing proliferation, angiogenesis, and metastasis, and reducing the effectiveness of immune regulation and the effects of chemotherapeutic agents [44,45]. Preoperative inflammation seems to aggravate tumors and negatively affect the prognosis of cancer [46,47,48,49,50,51,52]. Han et al. [50] reported that the presence of preoperative inflammation is an independent prognostic factor for a poor 3-year overall survival (33% vs. 73%, *p* = 0.001) in patients with gallbladder cancer. Cho et al. [52] found that the presence of preoperative inflammation is an independent prognostic factor for 3-year overall survival (21.5% vs. 66.1%, *p* = 0.003) and 3-year disease free survival (11.9% vs. 57.3%, *p* = 0.001) for patients with extrahepatic bile duct cancer. D’Silva et al. [49] described that preoperative c-reactive protein (CRP) > 1 mg/L was significantly associated with a poor overall survival (56.7% vs. 77.4%, *p* = 0.023). Saad MR et al. [47] also found that preoperative CRP was an independent poor prognostic factor for the overall survival and disease-free survival of patients with resected pancreatic ductal adenocarcinoma. The pathophysiologic mechanism of how acute inflammation affects cancer progression is now well-elucidated. One possible physiological mechanism is that inflammation promotes epithelial-to-mesenchymal transition (EMT). In this mechanism, the embryonic program loosens the cell–cell adhesion complexes and imparts to cell-enhanced migratory and invasive properties that cancer cells can select during metastatic progression [53,54,55,56]. Cancer cells that have undergone EMT are more aggressive and display increased invasiveness, a stem-like function, and resistance to apoptosis [57,58].

Based on the close relationship between inflammation and tumors, the use of anti-inflammatory agents, either alone or in combination may be a promising option for improving cancer prognosis. NSAIDs have long been studied as anti-cancer agents that can influence cancer development and progression [58]. A meta-analysis involving 300,000 colorectal, prostate, breast and lung cancer patients from 16 studies suggested that the potential NSAIDs could reduce distant metastasis in various types of cancer [59]. NSAIDs may be optimal candidates for cancer treatment, but unfortunately their use is limited by serious side effects. To reduce side effects among the NSAIDs, anti-inflammatory agents with more specific activity against cyclooxygenase 2 (COX-2) have been developed. Selective COX-2 antagonists (such as celecoxib) have a lower incidence of side effects compared to other NSAIDs, particularly gastrointestinal bleeding and dyspepsia [59,60,61]. Several preclinical studies evaluating the use of celecoxib in cancer therapy have shown the antitumor effects of these agents [62,63,64]. However, its anti-cancer properties have not been sufficiently established in clinical trials [61,65,66]. In the Randomized European Celecoxib Trial (REACT), patients showed no evidence of a disease-free survival benefit during 2 years of treatment with celecoxib compared to a placebo as an adjuvant treatment for ERBB2 (formerly HER2)-negative breast cancer [61]. In CALGB/SWOG 80702 (Alliance) clinical trial, the addition of celecoxib versus placebo for years to standard adjuvant chemotherapy in patient with stage III colon cancer did not significantly improve disease-free survival [65]. For these reasons, many anti-inflammatory agent can be used as adjuvants for conventional therapies, but further research is needed to better understand potential anticancer drugs.

## 5. Frailty

Biologic age is equal to physiological or functional age, whereas chronological age refers to physical or mathematical age. Although chronological age is not a modifiable risk factor, biologic age can be affected with prehabilitation. The recent biologic age model, which is gaining momentum, is based on frailty [66]. Frailty is a multidimensional phenotype that manifests through a variety of signs, symptoms, or other health-related problems. To define frailty as a clinical syndrome, the Fried frailty uses five phenotype criteria: (a) unintentional weight loss of ≥4.5 kg in the past year, (b) self-reported exhaustion, (c) muscle weakness (hand-grip strength in the lowest 20% quintile adjusted for sex and body mass index), (d) slow gait speed (the lowest quintile in walking time per 15 feet), and (e) low physical activity (the lowest quintile in kilocalories expended per week). Individuals meeting three or more of these criteria are considered frail [67]. Panayi AC et al. conducted a meta-analysis of 16 studies. They concluded that frailty is a prognostic indicator that strongly correlates with the risk of post-surgical morbidity and mortality [68]. They also suggested that prospective actions such as prehabilitation can improve the frailty status of surgical patients leading to better postoperative outcomes.

Prehabilitation is defined as the process of expanding patient’s functional and psychological ability to reduce the potentially detrimental effects of serious stressor [69]. Most prehabilitation programs consist of exercise, nutrition, and psychosocial components. The types of exercise interventions used range from walking to biking and stepping exercise and vary in duration and intensity [70].

## 6. Minimally Invasive Surgery

The first laparoscopic cholecystectomy was carried out in 1987 by Philippe Mouret and is considered the gold standard of abdominal surgery [71]. Laparoscopic surgery offers many benefits to patients, including shorter recovery and hospital stays, minimal postoperative pain, discomfort and disabilities, and improved cosmetic outcomes (less scarring) [72,73,74].

### 6.1. Surgical Stress Response after Minimally Invasive Surgery

Surgical injury is associated with oxidative stress, often due to ischemia/reperfusion injury [75]. Oxidative stress in critical illnesses is known to be associated with poor outcomes [76] and plays an important role in the development and manifestation of sepsis or systemic inflammatory response (SIRS). SIRS is a common cause of morbidity and mortality after surgery. The magnitude of this acute oxidative stress response reflects the severity of tissue trauma and surgery. Laparoscopic surgery has become popular due to its clinical advantages of minimized surgical trauma [77] and is associated with less systemic inflammation and preserved immune function [78]. A meta-analysis reported that laparoscopic surgery produces less systemic oxidative stress compared to open surgery [79].

### 6.2. Enhanced Recovery after Surgery (ERAS) Programs for Minimally Invasive Surgery

The concept of enhanced recovery after surgery (ERAS) was first introduced by Kehlet in the 1990 s and has been widely used in various surgeries over the past few decades [80,81,82]. The application of the ERAS protocol is expected to reduce surgical stress and complication rates by the combination of various interventions in the preoperative period [83]. It consists of 17 items, including preoperative consultation, optimal intraoperative epidural anesthesia, postoperative fluid restriction, early recovery mobilization, postoperative pain control, and early feeding [84,85]. Extensive research has shown that the ERAS protocol is superior to conventional surgical protocols [86,87,88]. Although, patients undergoing major abdominal surgery could benefit from minimally invasive surgery [89], some surgeons are still debating the benefits of ERAS when performing minimally invasive surgery. Thus, Li Z et al. [83] performed a systematic review and meta-analysis, including a comparative studies of the safety and efficacy of the ERAS protocol and conventional treatments for patients who underwent laparoscopic abdominal surgery. Previous studies showed that the ERAS protocol for laparoscopic abdominal surgery is safe and effective. In postoperative complications, the ERAS group was associated with lower overall postoperative complication rate. Furthermore, a subgroup analysis based on the study types (randomized clinical trials or non-randomized clinical trials) and surgery types (laparoscopic non-colorectal surgery or laparoscopic colorectal surgery) still proved that ERAS was beneficial with respect to lower postoperative complications when compared to laparoscopic surgery [89]. W R Spanjersberg et al. [90] also concluded that when laparoscopy and ERAS are combined, major morbidity and hospital stay were reduced. Since the reduction in morbidity appears to be due to a laparoscopic procedure rather than ERAS, laparoscopic itself may have independent benefits over ERAS treatment. 

## 7. Conclusions

Malnutrition, inflammation, frailty and surgical technique play crucial roles in the development of postoperative complications and patient prognosis. All surgical patients should be regularly screened for the presence or risk of malnutrition or inflammation before surgery. It should also be recognized that the surgical approaches may also affect the postoperative course. The identification of high-risk patients can trigger close surveillance after surgery. Additionally, the identification of high-risk patients can influence decision-making regarding treatment options, and clinicians should address the underlying causes of malnutrition and inflammation to promote patient survival.

## Figures and Tables

**Figure 1 nutrients-14-03107-f001:**
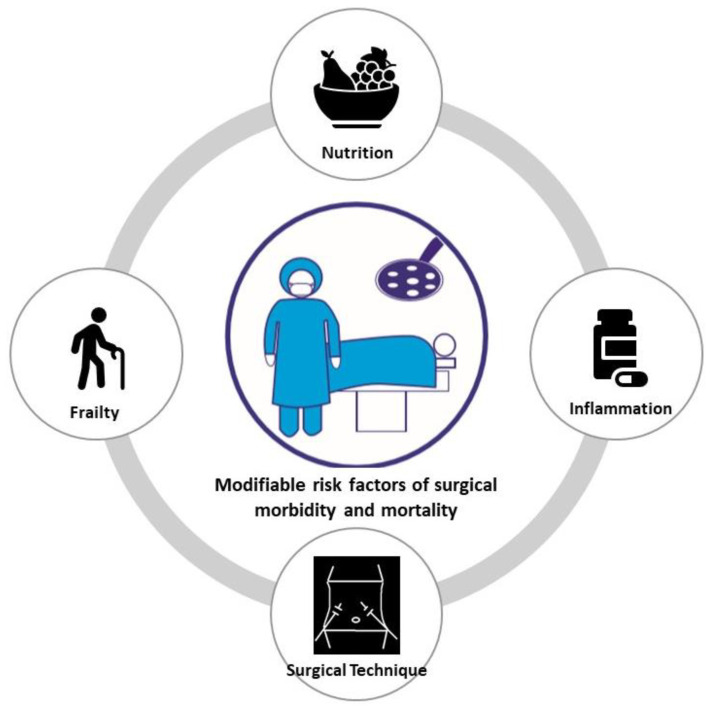
Summary of modifiable risk factors influencing postoperative outcomes.

## Data Availability

Not applicable.

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
