# Peer review of "Tackling Surgical Morbidity and Mortality through Modifiable Risk Factors in Cancer Patients"

_nutrients, 2022, doi:10.3390/nu14153107_

Round 1
Reviewer 1 Report
The manuscript, "Tackling Surgical Morbidity and Mortality through Modifiable Risk Factors", is a generally well written review of some of the modifiable factors that may be influenced to improve surgical outcomes in gastrointestinal cancer patients. I offer a number of suggestions:
1. The English used in the manuscript should be re-edited with particular attention to the use of articles (the, a, etc.) and to assure agreement of tenses (past, present, future) in sentences.
2. Are the authors able to present their methodology, namely how they chose the specific articles that they decided to reference from the vast array of possible studies and reviews?
3. The article and the references refer almost solely to issues in the context of gastrointestinal cancer patients. I suggest including the words "gastrointestinal" and "cancer" in the Title, the Introduction, and the Discussion/Conclusions.
4. In the Introduction, Frailty and Chronologic Age are not listed as sources of risk for cancer patients undergoing gastrointestinal surgery. Although Age is not a modifiable risk, Frailty may be influenced through prehabilitation. The authors might want to refer to age and frailty, and include a formal discussion of prehabilitation.
5. The manuscript's readers would benefit from a summary paragraph and graphic that lays out a clinical approach to the modification of surgical risk factors (e.g., elements of prehabilitation; possible perioperative use of anti-inflammatory agents (or not); inidcations for perioperative nutrition support; etc.).
Author Response
19th, July. 2022
Pf. Dr. Maria Luz Fernandez
Pf. Dr. Lluis Serra-Majem
Editor-in-Chief
Nutrients
On behalf of all co-authors, I would like to submit our revised manuscript titled “Tackling Surgical Morbidity and Mortality through Modifiable Risk Factors” for publication in this journal.
We appreciate the time and efforts that you have dedicated to provide insightful feedback to improve this manuscript. The suggestions helped us to see our article from a different perspective. We have revised our manuscript extensively as suggested, and have provided point-by-point responses to the reviewers’ comments. The changes made in response to the individual comments are described.
We sincerely hope that the manuscript is now suitable for publication in this journal, and would be pleased to respond to any further queries regarding this submission.
We look forward to hearing from you.
Sincerely,
Corresponding author: Ho-Seong Han
Department of Surgery, Seoul National University Bundang Hospital
82, Gumi-ro 173 Beon-gil, Bundang-gu, Gyeonggi-do 13620, Korea
Tel: +82-31-787-7091
Fax: +82-31-787-4055
E-mail: hanhs@snubh.org
Reviewer 1.
The manuscript, "Tackling Surgical Morbidity and Mortality through Modifiable Risk Factors", is a generally well written review of some of the modifiable factors that may be influenced to improve surgical outcomes in gastrointestinal cancer patients. I offer a number of suggestions:
- The English used in the manuscript should be re-edited with particular attention to the use of articles (the, a, etc.) and to assure agreement of tenses (past, present, future) in sentences.
à Thank you for your suggestions. We understand your suggestion. In order to improve readability, English has been corrected by expert team. The certification of proofreading is also attached.
- Are the authors able to present their methodology, namely how they chose the specific articles that they decided to reference from the vast array of possible studies and reviews?
à Thank you for your important comments. We added the methods section to show how we’ve chosen the specific article. We fully agree that the use of appropriate methodology in review articles is important in that readers are objectively up-to-date. In this aspect, presenting a methodology will help to increase the confidence of this review article. Thank you for providing a great way to improve the quality of our article.
After revision (Methods)
A systematic search of literature related to “nutrition” or “inflammation” or “minimally invasive surgery” and “cancer” in PubMed advanced search builder, Embase, web of science and Cochrane database up to May 2022 was conducted. The reference lists and all included studies were manually checked for further eligible studies. After exclusion of duplicates, the titles and abstracts of remaining records were screened independently by two authors (H-S Han, B Lee).
- The article and the references refer almost solely to issues in the context of gastrointestinal cancer patients. I suggest including the words "gastrointestinal" and "cancer" in the Title, the Introduction, and the Discussion/Conclusions.
à Thank you for important comments. We agree with the reviewer and have added the following words in the Title, introduction and discussion/conclusion section.
Title (Before revision)
Tackling Surgical Morbidity and Mortality through Modifiable Risk Factors
Title (After revision)
Tackling Surgical Morbidity and Mortality through Modifiable Risk Factors in Cancer patients
Abstract (Before revision)
Therefore, in this review article, we have described the current evidence regarding modifiable risk factors influencing surgical morbidity and mortality and future directions for improved postop-erative management.
Abstract (After revision)
Therefore, in this review article, we have described the current evidence regarding modifiable risk factors influencing surgical morbidity and mortality and future directions for improved postoperative management in cancer patients.
Introduction (Before revision)
Therefore, in this review article, we have summarized the current evidence regarding these modifiable risk factors influencing surgical morbidity and mortality.
Introduction (After revision)
Therefore, in this review article, we have summarized the current evidence regarding these modifiable risk factors influencing surgical morbidity and mortality in cancer patients.
- In the Introduction, Frailty and Chronologic Age are not listed as sources of risk for cancer patients undergoing gastrointestinal surgery. Although Age is not a modifiable risk, Frailty may be influenced through prehabilitation. The authors might want to refer to age and frailty, and include a formal discussion of prehabilitation.
à Thanks for raising these important points. We agree with you that frailty is an important modifiable risk factor for predicting postoperative morbidity and mortality. Therefore, we added the “Frailty” section.
After revision (Frailty)
Biologic age is equal to physiological or functional age, whereas chronological age is physical or mathematical age. Although chronological age is not a modifiable risk factor, biologic age can be affected with prehabilitation. Recently, the biologic age model that is gaining momentum is based on frailty [1]. Frailty is a multidimensional phenotype that manifest through a variety of sign, symptoms, or other health-related problems. To define frailty as a clinical syndrome, the Fried frailty uses five phenotype criteria: a) unin-tentional weight loss of ≥ 4.5 kg in the past year, b) self-reported exhaustion, c) muscle weakness (hand-grip strength in the lowest 20% quintile adjusted for sex and body mass index), d) slow gait speed (the lowest quintile in walking time per 15 feet), and e) low physical activity (the lowest quintile in kilocalories expended per week). Individuals meeting three or more of these criteria are considered frail [2]. Panayi AC et al. conducted meta-analysis of 16 studies. They concluded that the frailty is a prognostic indicator that strongly correlates with the risk of post-surgical morbidity and mortality [3]. They also suggested that prospective actions such as prehabiliation can improve the frailty status of surgical patients leading to better postoperative outcomes.
Prehabilitaion is defined as the process of expanding patient’s functional and psychological ability to reduce the potentially detrimental effects of serious stressor [4]. Most prehabiliation programs consist of exercise, nutrition, and psychosocial components. The types of exercise interventions used range from walking to biking and stepping exercise and vary in duration and intensity [5].
- Rockwood K, Andrew M, Mitnitski A. A comparison of two approaches to measuring frailty in elderly people. J Gerontol A Biol Sci Med Sci. 2007;62:738–43.
- Ji L, Jazwinski SM, Kim S. Frailty and Biological Age. Ann Geriatr Med Res. 2021 Sep;25(3):141-149.
- Panayi AC, Orkaby AR, Sakthivel D, Endo Y, Varon D, Roh D, Orgill DP, Neppl RL, Javedan H, Bhasin S, Sinha I. Impact of frailty on outcomes in surgical patients: A systematic review and meta-analysis. Am J Surg. 2019 Aug;218(2):393-400.
- Carli F, Zavorsky GS. Optimizing functional exercise capacity in the elderly surgical population. Curr Opin Clin Nutr Metab Care. 2005;8(1):23-32.
- Hijazi Y, Gondal U, Aziz O. A systematic review of prehabilitation programs in abdominal cancer surgery. Int J Surg. 2017 Mar;39:156-162.
- The manuscript's readers would benefit from a summary paragraph and graphic that lays out a clinical approach to the modification of surgical risk factors (e.g., elements of prehabilitation; possible perioperative use of anti-inflammatory agents (or not); inidcations for perioperative nutrition support; etc.).
- Thank you for your suggestions. As requested, we have added a figure to summarize this article.
Figure 1.

Reviewer 2 Report
Dear Authors
You have done great work and I think your results and conslusions are fine. I however need a Methods section in order to know how you selected and assessed and worked the publications used in this study.
Author Response
19th, July. 2022
Pf. Dr. Maria Luz Fernandez
Pf. Dr. Lluis Serra-Majem
Editor-in-Chief
Nutrients
On behalf of all co-authors, I would like to submit our revised manuscript titled “Tackling Surgical Morbidity and Mortality through Modifiable Risk Factors” for publication in this journal.
We appreciate the time and efforts that you have dedicated to provide insightful feedback to improve this manuscript. The suggestions helped us to see our article from a different perspective. We have revised our manuscript extensively as suggested, and have provided point-by-point responses to the reviewers’ comments. The changes made in response to the individual comments are described.
We sincerely hope that the manuscript is now suitable for publication in this journal, and would be pleased to respond to any further queries regarding this submission.
We look forward to hearing from you.
Sincerely,
Corresponding author: Ho-Seong Han
Department of Surgery, Seoul National University Bundang Hospital
82, Gumi-ro 173 Beon-gil, Bundang-gu, Gyeonggi-do 13620, Korea
Tel: +82-31-787-7091
Fax: +82-31-787-4055
E-mail: hanhs@snubh.org
Reviewer 2.
You have done great work and I think your results and conslusions are fine. I however need a Methods section in order to know how you selected and assessed and worked the publications used in this study.
à Thank you for your important comments. We also thanks for the positive comments and suggestions. Accordingly, we have added the methodology section. We fully agree that the use of appropriate methodology in review articles is important in that readers are objectively up-to-date. In this aspect, presenting a methodology will help to increase the confidence of this review article. Thank you for providing a great way to improve the quality of our article.
After revision (Methods)
A systematic search of literature related to “nutrition” or “inflammation” or “minimally invasive surgery” and “cancer” in PubMed advanced search builder, Embase, web of science and Cochrane database up to May 2022 was conducted. The reference lists and all included studies were manually checked for further eligible studies. After exclusion of duplicates, the titles and abstracts of remaining records were screened independently by two authors (H-S Han, B Lee).

This manuscript is a resubmission of an earlier submission. The following is a list of the peer review reports and author responses from that submission.